# Proteomic profiling reveals diagnostic signatures and pathogenic insights in multisystem inflammatory syndrome in children
Ulrikka Nygaard [1,2,11] ✉, Annelaura Bach Nielsen[3,11], Kia Hee Schultz Dungu [1,2], Lylia Drici [3], Mette Holm[4], Maud Eline Ottenheijm[3], Allan Bybeck Nielsen [1,5], Jonathan Peter Glenthøj[6], Lisbeth Samsø Schmidt[2,7], Dina Cortes[2,5], Inger Merete Jørgensen[2,6], Trine Hyrup Mogensen [8], Kjeld Schmiegelow[1,2], Matthias Mann[3,9], Nadja Hawwa Vissing[1,12] & Nicolai J. Wewer Albrechtsen [3,10,12]

Multisystem inflammatory syndrome in children (MIS-C) is a severe disease that emerged during the COVID-19 pandemic. Although recognized as an immune-mediated condition, the pathogenesis remains unresolved. Furthermore, the absence of a diagnostic test can lead to delayed immunotherapy. Using state-of-the-art mass-spectrometry proteomics, assisted by artificial intelligence (AI), we aimed to identify a diagnostic signature for MIS-C and to gain insights into disease mechanisms. We identified a highly specific 4-protein diagnostic signature in children with MIS-C. Furthermore, we identified seven clusters that differed between MIS-C and controls, indicating an interplay between apolipoproteins, immune response proteins, coagulation factors, platelet function, and the complement cascade. These intricate protein patterns indicated MIS-C as an immunometabolic condition with global hypercoagulability. Our findings emphasize the potential of AI-assisted proteomics as a powerful and unbiased tool for assessing disease pathogenesis and suggesting avenues for future interventions and impact on pediatric disease trajectories through early diagnosis.

Multisystem inflammatory syndrome in children (MIS-C) is a severe, life-threatening, immunological condition that occurs weeks after infection with SARS-CoV-2[1,2]. Although MIS-C is established as an immunological dys-regulation leading to cytokine storm[3], the underlying pathogenesis remains unresolved[4]. The incidence of MIS-C was approximately one in 3000 children infected with SARS-CoV-2 during the pre-Omicron waves, while the incidence decreased substantially as the Omicron variants became dominating[5–7]. This decrease has been attributed to a reduced ability of

Omicron to trigger hyperinflammation, as the Omicron variant is phylogenetically distinct from the pre-Omicron variants with enhanced immune escape[8]. Further, vaccination has been shown to decrease the incidence of MIS-C[9,10]. Still, sporadic MIS-C cases occur, and resurgence of MIS-C is possible with waning vaccine-induced immunity and novel variants of SARS-CoV-2.

Children with MIS-C present with fever and multiorgan involvement, including mucocutaneous, gastrointestinal, and cardiovascular

[1]Department of Pediatrics and Adolescent Medicine, Copenhagen University Hospital, Rigshospitalet, Copenhagen, Denmark. [2]Department of Clinical Medicine, University of Copenhagen, Copenhagen, Denmark. [3]NNF Center for Protein Research, Faculty of Health and Medical Sciences, University of Copenhagen, Copenhagen, Denmark. [4]Department of Pediatrics and Adolescent Medicine, Aarhus University Hospital, Aarhus, Denmark. [5]Department of Pediatrics and Adolescent Medicine, Copenhagen University Hospital Hvidovre, Hvidovre, Denmark. [6]Department of Pediatrics and Adolescent Medicine, Copenhagen University Hospital North Zealand, Hillerød, Denmark. [7]Department of Pediatrics and Adolescent Medicine, Copenhagen University Hospital Herlev, Herlev, Denmark. [8]Department of Infectious Diseases, Aarhus University Hospital, Aarhus, Denmark. [9]Proteomics and Signal Transduction, Max Planck Institute of Biochemistry, Martinsried, Germany. [10]Department of Clinical Biochemistry, Copenhagen University Hospital - Bispebjerg and Frederiksberg Hospital, Copenhagen, Denmark. [11]These authors contributed equally: Ulrikka Nygaard, Annelaura Bach Nielsen. [12]These authors jointly supervised this work: Nadja Hawwa Vissing, Nicolai J. Wewer Albrechtsen. ✉e-mail: Ulrikka.Nygaard@regionh.dk

involvement, often accompanied by circulatory shock[1,2,6]. The condition can be misinterpreted as sepsis, abdominal emergencies, and Kawasaki disease[11,12]. Thus, the lack of a diagnostic test for MIS-C can lead to delayed lifesaving immunomodulating therapy and prolonged unnecessary courses of broad-spectrum antibiotics.

To address this diagnostic challenge host-specific innovative omics methodologies have been suggested[4]. Proteomics can provide a comprehensive unbiased approach that investigates hundreds of plasma proteins simultaneously[13]. Proteomics has the potential to identify plasma proteins useful as diagnostic markers and explore disease mechanisms as circulating plasma proteins are markers of whole-body metabolic processes[14]. Due to recent technological improvements in proteomics pipelines, a comprehensive system-wide approach has become feasible[15]. However, a limitation of utilizing novel omics approaches is the interpretation of large amounts of complex data and the translation of this information into clinical medicine. This limitation may be surpassed by employing artificial intelligence (AI)-based techniques, which offer a powerful avenue for analyzing comprehensive unbiased proteomic data.

We employed AI-assisted proteomics to develop a unique diagnostic signature for children with MIS-C and to gain insight into the underlying disease mechanisms.

## Results

We enrolled 94 children, including 27 cases with MIS-C and 67 febrile controls consisting of 28 children with bacterial infection, 22 with viral infection, 7 with Kawasaki disease, and 10 with severe sepsis (Table 1, Fig. 1A). Children with MIS-C all had PCR-confirmed SARS-CoV-2 infection including 15 (56%) with the Alpha variant, 11 (41%) with the Delta variant, and one (4%) with the Omicron variant. None had comorbidities. Twenty-one of 27 (78%) presented with shock. Twenty-four (89%) were admitted to intensive care unit or semi-intensive care unit, and 9 (33%) received inotropes. Their blood samples for proteomics were collected before or within 24 h of treatment initiation. In nine of 27 (33%) patients, therapy with intravenous immunoglobulin was initiated before the blood sample was collected. Seven patients with MIS-C had additional samples for proteomics collected on days 2–4 after treatment initiation, and nine had additional samples collected when fully recovered a median of 39 days (19–76) following the diagnosis of MIS-C. All bacterial febrile controls had urinary tract infection confirmed by positive urine dipstick and urine culture with *Escherichia coli*. Viral controls had fever, proven viral detection in nasopharyngeal specimen, and C-reactive protein below 25 mg/L (Table 1). There were no deaths in any of the groups.

### Plasma proteins in children with MIS-C compared to controls

The data set used to identify differences in protein levels included patients with MIS-C, febrile controls with viral and bacterial infections, Kawasaki disease, and severe sepsis (Fig. 1a). We identified 450 plasma proteins across all plasma samples in the initial proteomic analysis (Fig. 1b, c). Three samples were excluded due to low numbers of measurable proteins, all from children with sepsis (Fig. 1d). After data quality assessment, 245 proteins were selected for further analysis. Further, 66 proteins related to therapy with intravenous immunoglobulin were excluded resulting in a total of 179 proteins in the final dataset (source data file: Supplementary Data 1). Overall, proteomic data separated disease categories as visualized by the uniform manifold approximation and projection (UMAP) plot (Fig. 2a) and the unsupervised heatmap (Fig. 2b), which both revealed a high correlation in children with MIS-C. A total of 105 proteins were significantly different in children with MIS-C compared to febrile controls, Kawasaki disease, and severe sepsis (Fig. 2c; Table 2). Figure 2d displays the overlap between significant proteins findings in MIS-C patients compared to the control groups combined or to each of the control groups separately (Supplementary Data 2).

Most proteins with significantly different levels between children with MIS-C and controls could be categorized into four groups: (1) Immunological response, (2) coagulation, (3) cell death and cell growth, and (4) lipid

profile (Table 2). *Immunological response*: Plasma proteins involved in the immunological response included elevated lymphocyte cytosolic protein 1 and Fc Gamma Receptor IIIa, both involved in adaptive immune response, as well as several elevated acute phase reactants including alpha-1-antichymotrypsin. Further, proteins playing a role in the innate immune response were significantly different in children with MIS-C, such as decreased levels of peptidoglycan recognition protein 2, and increased levels of several complement factors (Table 2). *Coagulation*: Numerous coagulation-related proteins differed significantly in children with MIS-C with reduced coagulation factors XII and XIII, increased procoagulants fibrinogen and Von Willebrand Factor, and reduced anticoagulants, among others antithrombin, protein C, and platelet factor 4. Children with MIS-C also had different levels of proteins related to the recruitment and activation of platelets. *Cell death and growth*: The levels of actin B, extracellular matrix protein 1, fibronectin, and other proteins implicated in cell and tissue remodeling were affected in MIS-C. *Lipid profile*: Finally, the lipid profile in children with MIS-C was different from febrile controls with reduced apolipoproteins A, C, and H, and elevated apolipoproteins E and F.

Unsupervised protein-protein co-expression network analyses, guided by machine learning, revealed eight clusters of proteins (Fig. 2e). These co-expression clusters elucidated interactions between apolipoproteins and proteins involved in the immune response (clusters 0, 1, and 2), proteins participating in the complement cascade (cluster 3), proteins involved in coagulation (cluster 4), proteins playing a role in oxygen transport (cluster 6), and proteins impacting platelet function (cluster 7). Cluster 5 was composed of heterogeneous proteins related to coagulation, inflammation, and liver function. The proteins that differed significantly in children with MIS-C were explored by unbiased pathway enrichment analyses and revealed 15 biological pathways, also primarily involving (1) immunological responses, (2) coagulation, (3) cell death and cell growth, and (4) platelet activation (Fig. 2f).

### Diagnostic classifier for MIS-C using machine learning

The data set used to develop a diagnostic signature for MIS-C included children with MIS-C and febrile controls with viral and bacterial infections (Fig. 3a). All 12 machine-learning algorithms, except one, had Matthews Correlation Coefficient (MCC) and area under the curve (AUC) between 0.77-1 (Fig. 3b). We continued the subsequent analysis with the support vector classification (SVC) model that had an AUC and MCC of 100% and 1, respectively (Fig. 3b). Recursive feature elimination revealed that only four proteins were necessary to obtain a high predictive performance (Fig. 3c). The four selected proteins were lymphocyte cytosolic protein 1, Fc Gamma Receptor IIIa, alpha-1-antichymotrypsin and butyrylcholinesterase (Table 2; Fig. 3e, Supplementary Data 3).

### Validation of the 4-protein diagnostic signature

When the 4-protein diagnostic signature was applied to the test set, an AUC of 100% was achieved (Fig. 3f). The median prediction probability was 83.0% (IQR 11.8) for the patients with MIS-C at the acute stage, 9.1% (IQR 6.0) for viral infections, and 12.2% (IQR 7.1) for bacterial infections (Fig. 3d, Supplementary Data 4). When applying the 4-protein diagnostic signature on the internal validation cohorts, including MIS-C patients on days 2–4 of treatment initiation, fully recovered MIS-C patients, Kawasaki disease and severe sepsis, the combined AUC was 93.4% (95% CI 92.1–94.7). Children with MIS-C, who had received immunomodulating therapy for 2–4 days, had a median prediction probability of 87.7% (IQR 9.3), while children who had recovered fully after MIS-C had a median MIS-C prediction probability of 7.3% (IQR 1.5) (Fig. 3d, Supplementary Data 4). Children with severe sepsis and Kawasaki disease had a median prediction probability of 20.7% (IQR 13.1) and 55.8% (IQR 43.7), respectively.

To evaluate the generalizability of AI-based proteomic prediction of MIS-C, an external U.S. validation cohort of 25 children with MIS-C and 34 healthy controls was investigated (Fig. 3a). As the proteins used in our 4-protein signature were not measured in the external cohort, the 28 plasma proteins measured in both studies were used to assess the validity of our AI-

**Table 1 | Characteristics of patients with MIS-C and febrile controls**

| | MIS-C | | | Febrile controls | | Kawasaki disease | Severe sepsis[b] |
|---|---|---|---|---|---|---|---|
| | Acute stage | During admission | Full recovery | Bacterial[a] | Viral[b] | | |
| No. of patients | 27 | 7 | 9 | 28 | 22 | 7 | 10 |
| Sex (males/females) | 14/13 | 3/4 | 3/6 | 3/25 | 10/12 | 6/1 | 5/5 |
| Age, years | 9 (5–15) | - | - | 9 (4–15) | 11 (6–15) | 2 (1–4) | 4 (1–11) |
| Blood sample collection (day[d]) | 0 (-2–0) | 3 (2–4) | 39 (19–76) | 0 (0–5) | 0 (0–1) | 1 (0–18) | 0 (0–2) |
| **Clinical characteristics[e]** | | | | | | | |
| Hypotension | 21 (78%) | 0 | 0 | 0 | 0 | 0 | 3 (30%) |
| Cardiac involvement | 27 (100%) | - | - | 0 | 0 | 0 | 1 (10%) |
| Gastrointestinal involvement | 25 (93%) | - | - | 0 | 0 | 0 | 2 (20%) |
| Hematologic involvement | 27 (100%) | - | - | 0 | 0 | 0 | 1 (10%) |
| Dermatologic involvement | 25 (93%) | - | - | 0 | 0 | 7 (100%) | 3 (30%) |
| Neurologic involvement | 0 | - | - | 0 | 0 | 0 | 2 (20%) |
| Respiratory involvement | 7 (26%) | - | - | 0 | 0 | 0 | 5 (50%) |
| Renal involvement | 13 (48%) | - | - | 0 | 0 | 0 | 0 |
| **C-reactive protein** (mg/L, max) | 219 (84-402) | 87 (83-202) | <10 | 110 (44–307) | 5 (0–13) | 169 (50–324) | 285 (194–356) |
| **Treatment** | | | | | | | |
| Antibiotic therapy | 26 (96%) | - | - | 28 (100%) | 5 (23%) | 7 (100%) | 12 (100%) |
| Inotropes | 9 (33%) | | | | | | 2 (20%) |
| Immunoglobulins | 24 (89%) | - | - | - | - | 7 (100%) | 0 (0%) |
| Glucocorticoids | 24 (89%) | - | - | - | - | 2 (29%) | 0 (0%) |
| Anakinra | 10 (37%) | - | - | - | - | 0 (0%) | 0 (0%) |
| **Intensive Care Unit** | 24 (89%) | | | | | | 4 (40%) |
| **Length of hospital stay** (days, range) | 6 (3-13) | - | - | 1 (0–9) | 0 (0–4) | 6 (3–14) | 8 (2–21) |

Data are given as median (ranges) or proportions.

[a] Bacterial infection included children with proven urinary tract infections

[b] Viral controls had fever, proven viral detection in nasopharyngeal specimen, and C-reactive protein below 25 mg/L

[c] Children with severe sepsis had blood cultures with *Neisseria meningitidis* (*N* = 2), Group A *Streptococcus* (*N* = 1), *S aureus* (*N* = 1), and Group B *Streptococcus* (*N* = 2), and *H. influenzae* meningitis (*N* = 1) and perforated appendicitis (*N* = 2) (*E coli* and *Klebsiella*)

[d] The day following treatment initiation (MIS-C) or hospital admission (febrile controls), where the blood sample was collected.

[e] Organ system involvement was defined according to the CDC with the following criteria: (1) Cardiac involvement, e.g. elevated troponin and/or N-terminal pro B-type natriuretic peptide, abnormal echocardiogram, or arrhythmia; (2) respiratory involvement, e.g. pneumonia, acute respiratory distress syndrome, or pulmonary embolism; (3) renal involvement, i.e. acute kidney injury or renal failure; (4) gastrointestinal involvement, e.g. abdominal pain, vomiting, diarrhea, elevated liver enzymes, ileus, gastrointestinal bleeding; (5) neurologic involvement, i.e. seizure, stroke or aseptic meningitis; (6) hematologic involvement, i.e. thrombophilia or thrombocytopenia, elevated D-dimers, and/or (7) dermatologic involvement, e.g. erythroderma, mucositis, other rashes. Shock was defined as persistent blood pressure below the five percentile, according to age

based approach for MIS-C diagnostics. The new support vector classification model including the 28 proteins had a high prediction performance with an AUC of 86.7% (95% CI 79.7-93.7) (Fig. 3f).

## Discussion

In this study, we employed AI to complex proteomics data to develop a diagnostic signature for children with MIS-C and explore the underlying biological mechanisms of the disease. The performance of multiple machine learning algorithms revealed that MIS-C could be discriminated from children with bacterial and viral infections as we identified a highly accurate diagnostic signature with an AUC of 100% based on only four proteins. The 4-protein diagnostic signature holds promising avenues for developing a rapid, and low-cost, diagnostic bedside test with important implications for early recognition and targeted treatment. Further, we found proteomics to be a powerful and unbiased tool for assessing disease pathogenesis in children with MIS-C. AI could extract intricate protein patterns, beyond the reach of traditional methods, which indicated MIS-C as a condition with immune dysregulation closely linked to changes in apolipoproteins, global hypercoagulability, and high cell and tissue remodeling.

The diagnosis of MIS-C is based on clinical manifestations and elevated acute phase reactants, such as C-reactive protein, which are often indistinguishable from a wide range of other diseases[11,12]. The lack of a diagnostic test has resulted in delays with targeted immunomodulating treatment and

unnecessary courses of broad-spectrum antibiotics. The diagnostic signature found in this study was based on only four plasma proteins, of which three were involved in the immune response. Applying the machine learning technique, recursive feature selection, revealed that these four plasma proteins, among a total of 179, were sufficient to differentiate MIS-C from other febrile conditions. Few patients with septic shock and Kawasaki disease were overlapping, reflecting the possible shared pathophysiological features between these conditions. The 4-protein diagnostic signature had high accuracy in children with MIS-C 2–4 days following initiation of immunomodulating therapy. This demonstrates its robustness to delayed sample collection and partial clinical recovery. While we could not validate our 4-protein signature on the external U.S. validation cohort (as those four proteins were not part of their protein panel), we successfully demonstrated the validity of our AI-based approach for MIS-C diagnostics, as the new support vector classification model, using different proteins, also achieved a high diagnostic accuracy of MIS-C. During our algorithm selection, we also found that several models, including different proteins, had high AUCs. This supports that several proteins may be used for an MIS-C signature and emphasizes proteomics as a very powerful tool for MIS-C diagnostics.

Previously, a 3-protein signature has been shown to distinguish MIS-C patients from other disease controls with an AUC of 86% in a study investigating seven host proteins[16]. Additionally a diagnostic signature based on a 5-gene blood RNA expression signature for MIS-C has been

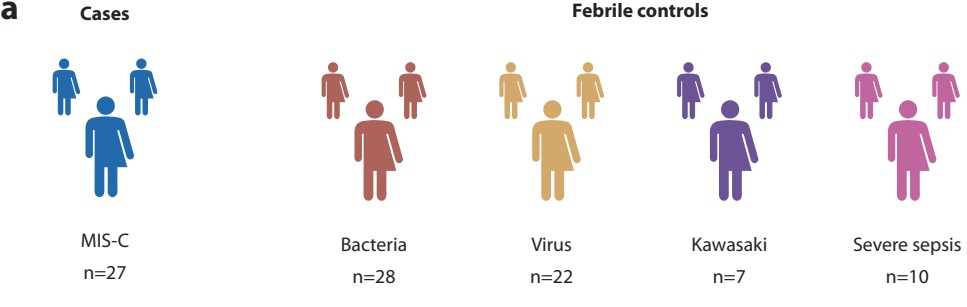

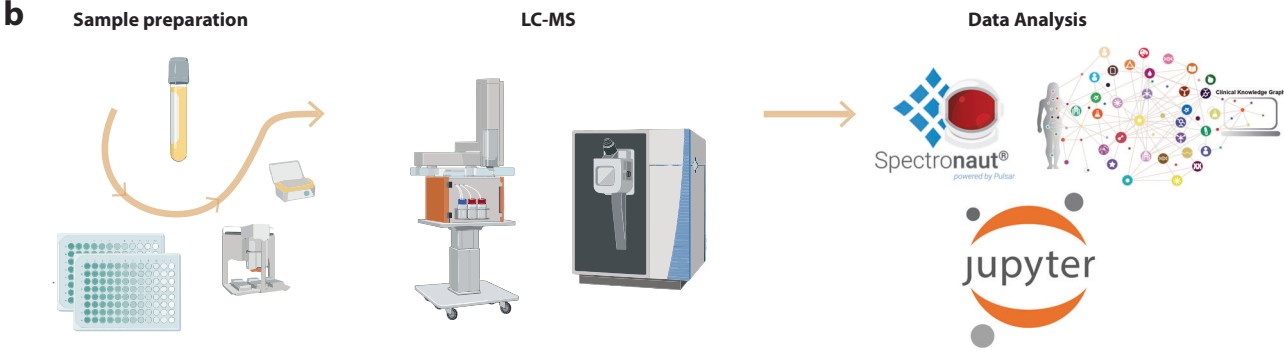

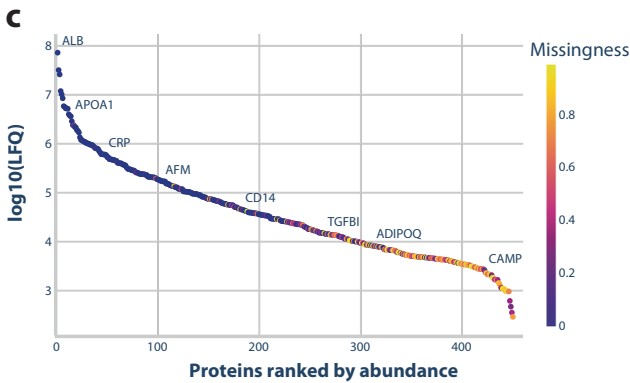

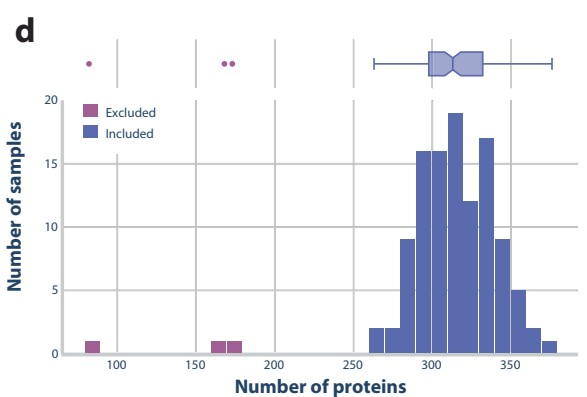

**Fig. 1 | Patient overview and proteomics workflow. a** Overview of the number of samples included in our study distributed on different diagnostic groups. **b** Laboratory and analytical workflow involving sample preparation of plasma using semi-automated BRAVO robot as well as liquid chromatography using Evosep One system and tandem mass spectrometry by Exploris 480 Thermo Fischer Scientific system. Data analysis was performed by Spectronaut processing of raw mass-spectrometry data followed by bioinformatic analysis with the Clinical Knowledge Graph in a Jupyter Python environment. Created with Adobe Illustrator software

and Biorender.com. **c** Dynamic range of the 450 proteins measured by liquid chromatography. Proteins are ranked by abundance and colored by missingness across all samples. Named proteins illustrate the concentration range measured; from highest abundant (albumin, ALB) to low abundant signaling molecules (cathelicidin antimicrobial peptide, CAMP; adiponectin, ADIPOQ). **d** Histogram of the number of proteins measured in each sample. Three samples were excluded from further analysis due to low protein numbers (marked in pink).

reported[17]. It was validated with an RT-qPCR assay and revealed a high diagnostic accuracy. Collectively, these results suggest that MIS-C has distinct host responses detectable by both transcriptomics and proteomics, which may be suitable for innovative diagnostic tests. However, a diagnostic test based on few plasma proteins may reveal a faster result and may be cheaper to implement in clinical medicine as a routine laboratory analysis.

The pathogenesis of MIS-C was explored using the extensive proteome dataset obtained by unbiased mass spectrometry, which revealed global changes in mechanisms and pathways involved in the pathogenesis of MIS-C. Immune dysregulation was indicated by the increased levels of proteins in multiple pathways leading to hyperinflammation, including both the innate and adaptive immune response, as described in previous studies[18–20]. Consistently, three of the four proteins in the identified 4-protein signature, were

involved in immune dysregulation: Alpha-1-antichymotrypsin is involved in complement activation, while Fc Gamma Receptor IIIa is involved in antibody-dependent cellular toxicity. Further, lymphocyte cytosolic protein 1, which was elevated, indicated activation of T-cells, supporting the hypothesis of a super-antigen-mediated polyclonal T-cell activation[21]. The last protein in the 4-protein signature was butyrylcholinesterase, which hydrolyzes choline esters and was reduced. The significance of this protein in the pathogenesis of MIS-C is unknown.

We found profound changes in lipid metabolism, consistent with previous studies[22,23] Lipid mediators have been described to be involved in vasodilation and increased vascular permeability[24], a frequent and severe clinical manifestation of MIS-C[6]. Further, the unsupervised protein-protein co-expression network analyses conducted by machine learning indicated

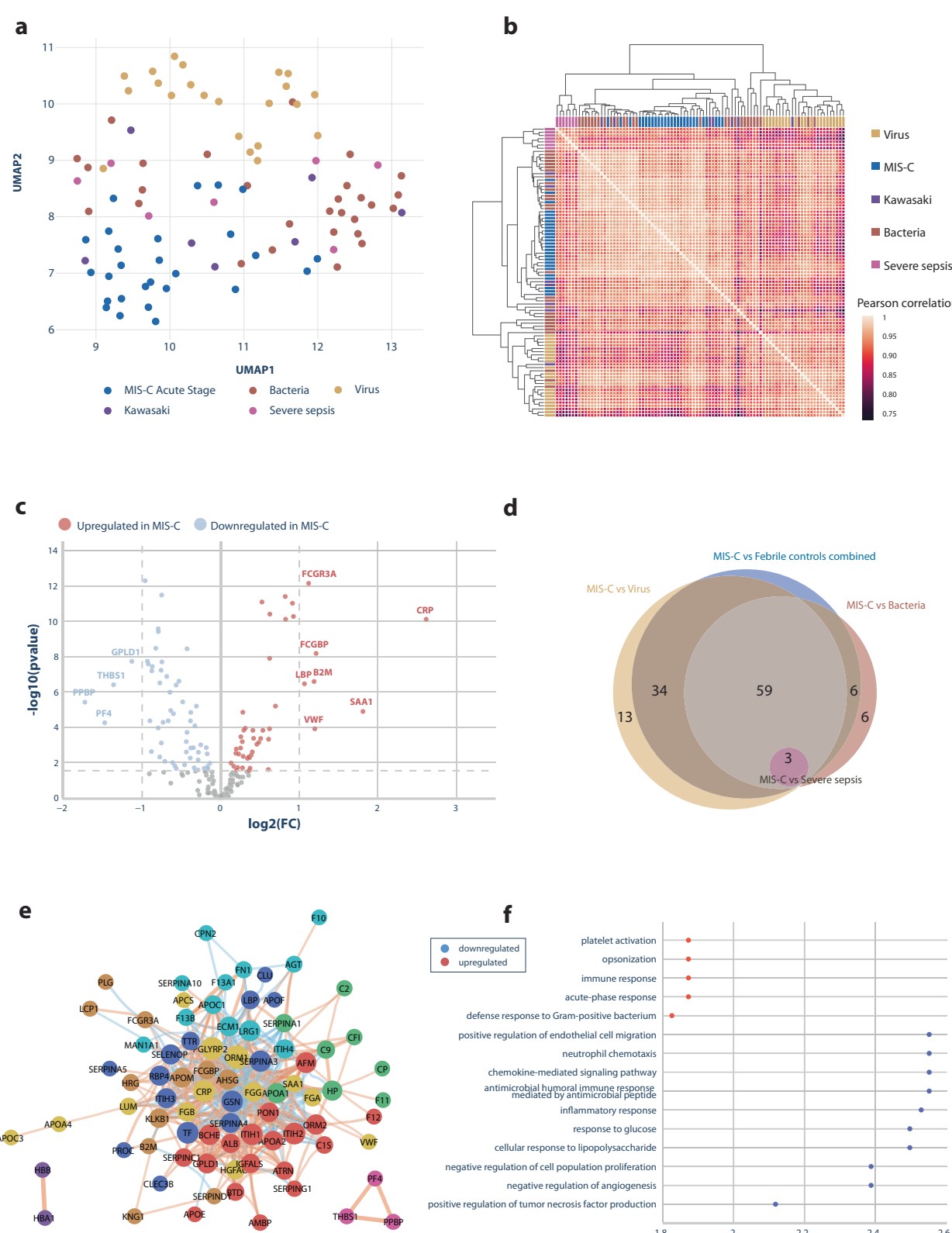

an intricate connection between apolipoproteins and immune dysregulation, which may be due to the function of lipids as proinflammatory mediators, as described in other conditions[25]. It suggests MIS-C as an immunometabolic condition[26]. Alterations in numerous coagulation factors with elevated procoagulants, reduced anticoagulants, and impaired fibrinolysis point towards a global hypercoagulability and may explain the risk of thrombosis in children with MIS-C[27]. Overall, our results align with those of a previous study, which also reported upregulated Fc Gamma Receptor IIIa, immune and complement activation and reduced lipid transport and clearance mechanisms[28]. Furthermore, our results were

**Fig. 2 | Differences in protein levels between MIS-C and controls. a** Uniform manifold approximation and projection (UMAP) analysis of proteomic data from each patient. The diagnostic groups are shown by different colors. **b** An unsupervised heatmap of Pearson correlations between each pair of proteome samples. The color scale represents Pearson correlation values with light colors indicating values close to 1 and dark values close to 0.75. Hierarchical clustering of the samples is shown on the top and left side of the heatmap, where the diagnostic group of each sample is represented by different colors. **c** A volcano plot showing proteomic differences between MIS-C and febrile controls. Each point represents a protein. The x-axis represents log2 fold change, and the y-axis represents the log10 (p-value). The dashed line represents the statistical significance threshold after adjusting for multiple testing. Proteins that were increased in MIS-C patients compared to febrile controls are shown in red, such as C-reactive protein and Fc Gamma Receptor IIIa (FCGR3A). Proteins that were decreased in MIS-C are shown in blue, including platelet factor 4 (PF4) and pro-platelet basic protein (PPBP). Proteins that did not differ significantly in MIS-C are shown in gray. Proteins marked with a name were both significant and had a high log2 fold change (>1 or <−1). **d** A Venn diagram illustrating the overlap of proteins that were statistically different when MIS-C protein levels were compared to those of each febrile condition or febrile controls combined. For example, 59 significant proteins overlapped between all comparisons except severe sepsis. No proteins differed between MIS-C and Kawasaki disease after adjusting for multiple testing. **e** Protein-protein network based on the proteins with significantly different levels in MIS-C compared to febrile controls. The network is based on the Spearman correlation of protein levels followed by Louvain clustering. As an example, cluster 7 illustrates three platelet-related proteins, platelet factor 4 (PF4), pro-platelet basic protein (PPBP), and thrombospondin-1 (THBS1), with significantly lower protein levels in MIS-C patients compared to febrile controls. **f** Gene ontology biological processes enrichment analysis of proteome data identified 15 biological pathways affected in MIS-C patients compared to febrile controls. The red color illustrates an upregulated pathway and the blue a downregulated pathway. The x-axis shows the order of statistical strength by adjusted p-value to the log10.

consistent with studies exploring single coagulation factors or coagulation profiles[29,30]. The significant alterations in proteins related to cell growth, cell death, and/or cell remodeling in children with MIS-C are unprecedented but align with the profound dysregulation of cellular and immunological processes and the multiorgan nature of the disease[3,6]. These results serve as a proof-of-concept of AI-assisted proteomics in exploring disease mechanisms of new diseases, or complex diseases not yet fully comprehended.

This study has several limitations. First, while the quality and size of the dataset were sufficient to develop a diagnostic signature, the size of our validation cohort was too small to identify significant protein changes between children with MIS-C and Kawasaki disease. Second, there is a possibility of spurious findings, unrelated to MIS-C, in our large proteome dataset, as we were unable to explain the function of all significantly altered proteins. As AI techniques model complex data patterns, understanding the precise features influencing the predictions can be challenging. Nevertheless, the disease mechanisms we discovered are supported by clinical correlations and the existing knowledge of the disease pathogenesis and pathophysiology. Third, we were unable to validate the 4-protein signature from external validation cohorts, as studies including those four proteins have not been published. Further, the decline in the incidence of MIS-C during the Omicron era refrained us from validating the accuracy of the 4-protein signature in a prospective MIS-C cohort. Finally, the MIS-C decline could challenge the relevance of our findings. However, as a resurgence of MIS-C remains possible with new variants and waning vaccine-induced immunity, we find that continuous investigation of this severe and potentially life-threatening disease is important.

In conclusion, we harnessed the power of AI to explore complex proteomic data from children with MIS-C, a condition that remains elusive. The study demonstrated the potential of proteomics to impact pediatric disease trajectories through early diagnosis as we identified a 4-protein diagnostic signature that was accurate in distinguishing MIS-C from children with phenotypically similar diseases. We provided a global characterization of proteomic changes in the pathogenesis of MIS-C, emphasizing AI-assisted proteomics as a powerful and unbiased tool for assessing disease pathogenesis and potentially paving the way for more efficient future interventions.

## Methods
This nationwide population-based study prospectively included patients aged 0–17 years with MIS-C from all 18 Danish pediatric departments from April 1, 2020 to March 15, 2022[6,10]. Patients met the US Centers for Disease Control and Prevention MIS-C case definition. Febrile controls consisted of children with viral and bacterial infections. Patients with Kawasaki, and severe sepsis were enrolled from January 1, 2019, to December 31, 2019, before the COVID-19 pandemic. Children with Kawasaki disease met American Heart Association criteria for complete or incomplete disease. Two pediatric infectious disease specialists adjudicated the final diagnosis for all patients when testing results and clinical outcomes were known.

Sample size calculations were not performed due to the exploratory nature of the study.

Patients were recruited under approval by the research ethics committees of the Ethics Committee of Capital Region of Denmark (H-20028631) and the Danish Data Protection Agency (P-2019-29). Informed oral and written parental consent was provided before participation. All ethical regulations relevant to human research participants were followed. The study was registered at ClinicalTrials.gov, NCT05334134.

### Liquid chromatography mass spectrometry data analysis
Venous blood samples were collected into EDTA-containing tubes, spun at 3000 g for 10 min at 4 °C within 2 h, and stored at -80 °C. Sample preparation for proteomic analysis was performed as previously published[31]. Samples were analyzed using an Exploris 480 Thermo Fischer Scientific system by Evosep One (Evosep Biosystem, Denmark) and proteomic data were acquired in a data-independent acquisition mode. Proteins related to therapy with intravenous immunoglobulin, including heavy-chains, light-chains, j-chains and variable regions, were excluded[32]. The mass spectrometry raw files were processed with Spectronaut version 17 (Biognosys, Zurich, Switzerland). A previously generated plasma spectral library containing 2137 protein groups and 16,254 peptides was used.

### Statistics and reproducibility
Data was processed using the Clinical Knowledge Graph and Jupyter Notebook[33]. In short, protein intensities were log-transformed, a stringent filter for missingness was applied (>70% completeness across all samples and at least 50% within each group), and missing values were imputed based on a downshifted normal distribution. Sample quality was assessed as described previously[34]. Batch correction was performed with Clinical Knowledge Graph (pyCombat) to ensure that the plate a sample was run on would not affect downstream results. Unpaired t-tests were used to identify proteins with significantly different levels between the cohorts. Multiple hypothesis correction was applied using the Benjamini-Hochberg method, with adjusted $P$-values < 0.05 considered statistically significant.

UMAP was performed to illustrate the underlying structure of the data. The UMAP plot reduces the multidimensional proteomic data into two dimensions, thereby allowing the separation of patient group by visual interpretation. Hierarchical clustering using Pearson correlation distance was used to compute a sample correlation heatmap. Both the UMAP and heatmap of Pearson correlations were unsupervised, meaning that the grouping of individuals was based on the proteomic data alone and not informed by disease grouping. Volcano plots were used to visualize plasma proteins that differed significantly between MIS-C and controls. Gene Ontology Biological Process enrichment analysis was performed to identify the enrichment of biological processes based on a set of significantly different proteins. Protein-protein co-expression clusters were identified with Clinical Knowledge Graph by Spearman correlation analysis, followed by Louvain network clustering. The clusters were visualized using Cytoscape[35].

**Table 2 | Plasma protein alterations in children with MIS-C**

| Protein name | Protein symbol | Level | Protein description |
|---|---|---|---|
| **Immune response** | | | |
| **Adaptive immune response** | | | |
| Lymphocyte Cytosolic Protein 1[a] | LCP1 | ↑ | **Activates T-cells in response to co-stimulation through TCR/CD3 and CD2 or CD28.** |
| Fc Gamma Receptor IIIa[a] | FCGR3A (CD16a) | ↑ | Cell surface receptor that binds antigen-IgG complexes and triggers antibody-dependent cellular cytotoxicity |
| Beta-2-microglobulin | B2M | ↑ | A component of the MHC class I molecule, which presents antigen to cytotoxic T cells |
| Fc γ Binding Protein | FCGBP | ↑ | Interacts with the Fc portion of immunoglobulin G |
| **Acute phase reactants** | | | |
| Alpha-1-antichymotrypsin[a], C-reactive protein, orosomucoid, serum amyloid, haptoglobin | SERPINA3 CRP, ORM1/2, SAA1, HP | ↑ | Acute phase reactants promote, among others, inflammation, facilitate phagocytosis, and activate the complement system |
| **Innate immune response** | | | |
| Peptidoglycan Recognition Protein 2 | PGLYRP2 | ↓ | Recognizes peptidoglycan and triggers the release of pro-inflammatory molecules |
| Glycoprotein CD14 | CD14 | ↑ | CD14 is a co-receptor for Toll-like receptors (TLRs), which increases pro-inflammatory cytokines, such as tumor necrosis factor-alpha (TNF-α) and interleukin-1 beta (IL-1β) |
| Complement factors | C1QB, C1S, C2, C4BPB, C4B, C9, CFI | ↑ | Activation of the classical complement pathway leads to a widespread inflammatory response |
| Ficolin 2 | FCN2 | ↑ | Alternative complement factors |
| **Coagulation** | | | |
| Coagulation factors | F12, F13A1, F13B | ↓ | Procoagulants: F12 initiates coagulation and F13 converts fibrinogen to fibrin |
| | F10, F11 | ↑ | Procoagulants: Key components in the coagulation cascade |
| Fibrinogen | FG | ↑ | Procoagulant: When converted to fibrin by thrombin |
| Von Willebrand Factor | vWF | ↑ | Procoagulant: Aggregates platelets and activates FVIII |
| Antithrombin | SERPINC1 | ↓ | Anticoagulant. Inhibits thrombin and several coagulation factors |
| Protein C | PROC | ↓ | Anticoagulant: Inhibits FV & FVIII, and thrombin |
| Heparin Cofactor II | SERPIND1 | ↓ | Anticoagulant: Inhibits thrombin |
| Platelet factor 4 | PF4 | ↓ | Anticoagulant: Neutralizes heparin |
| Kallikrein | KLKB1 | ↓ | Anticoagulant: Activates plasminogen to plasmin through bradykinin |
| Plasminogen | PLG | ↓ | Initiates fibrinolysis (dissolution of thrombus) |
| Thrombospondin-1 | THBS1 | ↓ | Initiates platelet activation and aggregation |
| Pro-platelet basic protein | PPBP | ↓ | Recruit other platelets |
| **Cell death and cell growth** | | | |
| Actin B | ACTB | ↓ | A key component of the cytoskeleton |
| Extracellular matrix protein 1 | ECM1 | ↓ | Participates in tissue repair and angiogenesis |
| Fibronectin | FN1 | ↓ | Facilitates tissue remodeling |
| Clusterin | CLU | ↓ | Involved in tissue remodeling and has anti-apoptotic effects |
| **Lipid metabolism** | | | |
| Apolipoproteins | APOE, APOF | ↑ | The major component of high-density lipoprotein |
| Apolipoproteins | APOA, APOC1, APOC3, APOH | ↓ | Components of high-density lipoprotein (HDL) and very-low-density lipoprotein (VLDL) |
| **Others** | | | |
| Butyrylcholinesterase[a] | BCHE | ↓ | Hydrolyzes choline esters |

[a]Proteins in the 4-protein diagnostic signature

↓ Indicate reduced protein levels in MIS-C compared to controls; ↑ indicates increased protein levels in MIS-C compared to controls.

Supervised machine learning analysis was performed to investigate the feasibility of a diagnostic signature for MIS-C. Our dataset including children with MIS-C and febrile controls with viral and bacterial infections was divided into a training set (80%) for model development and a test set (20%) for model validation in a 5-fold cross-validation scheme. Each set had the same ratio of MIS-C and febrile control samples. Z-scored data (mean 0 and standard deviation 1 within each sample) was used as input. MIS-C diagnosis (yes/no) was used as the classification target. Twelve different machine-learning algorithms were investigated. A final model was based on a hyperparameter search (random grid search specific to each algorithm) and recursive feature elimination (only for models with a feature importance attribute) combined with 5-fold cross-validation to identify the balance point between a minimal combination of proteins and a high predictive performance. Model selection was based on the highest prediction

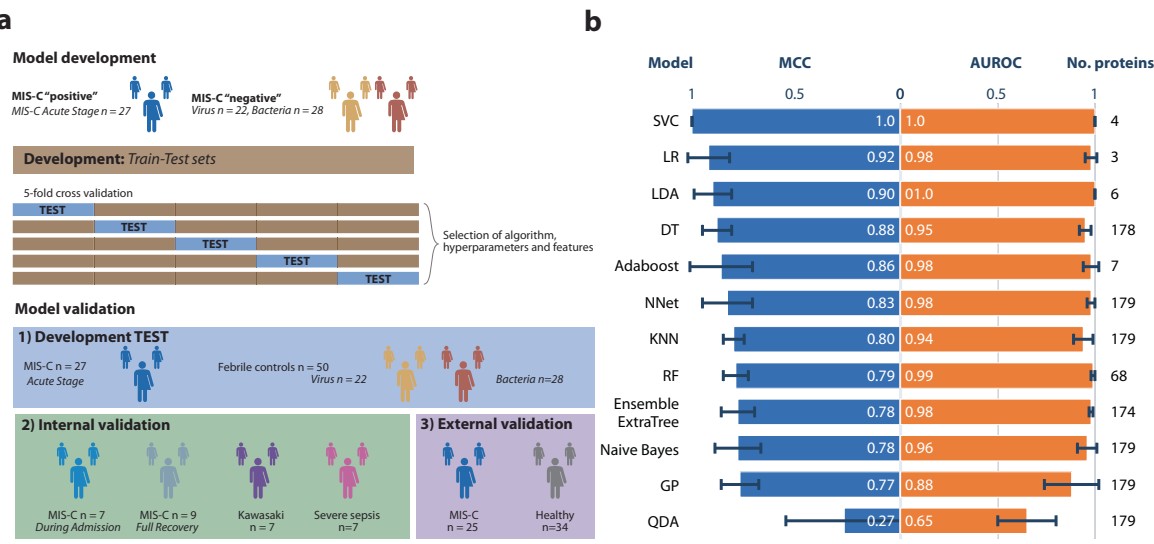

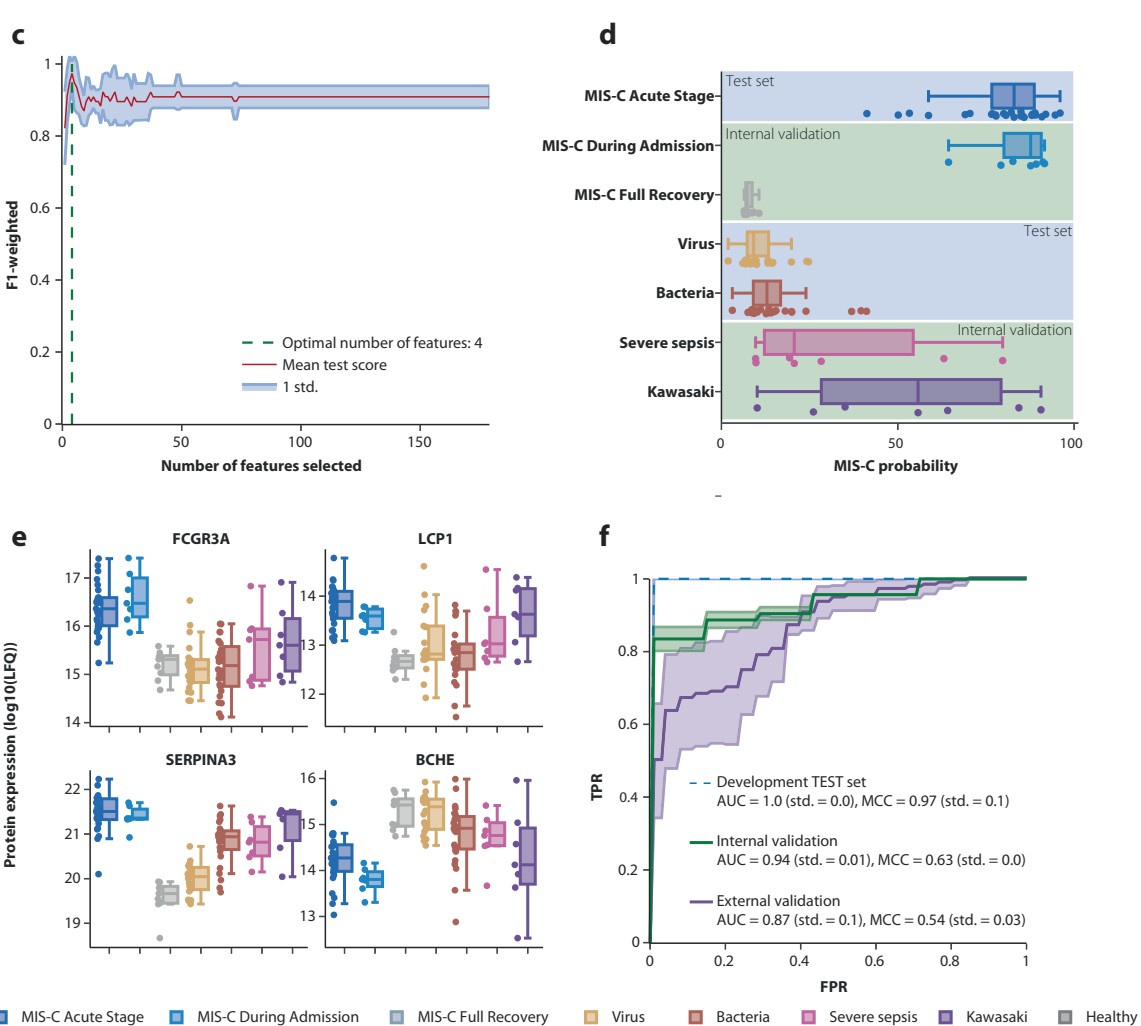

performance using the area under the receiver operating characteristic curve, AUC and MCC. Prediction probabilities were calibrated for the final model. Machine learning analyses and model calibration were performed using Python (3.7.9) in combination with the scikit-learn (sklearn) library[36]. Analyses were performed using the scikit-learn (sklearn) Python library.

The performance of the developed model was tested on both the test set and internal validation cohorts including Kawasaki disease, severe sepsis, and children with MIS-C, both during partial and full recovery. The model was also tested on an external validation cohort from the Proteomics Identifications Database (PRIDE) consisting of 22 children with MIS-C and

**Fig. 3 | Development of a diagnostic signature using artificial intelligence.**
**a** Overview of the strategy used for machine learning analysis. Model development was carried out on samples from patients with MIS-C (acute stage), virus, and bacteria. Model development was split in 5 cross-validation folds with 80% in the training set and 20% in the test set. Cross-validation was used to assess which combination of machine learning algorithms that resulted in the most optimal model. After model training and selection, model performance was reported on the test set from model development, on internally collected validation cohorts, including Kawasaki disease, severe sepsis, MIS-C 2–4 days after treatment initiation ('during admission', i.e., still 'MIS-C positive') and after full recovery (MIS-C 'negative'). Lastly, the strategy of machine learning-based diagnostic support was applied to an external MIS-C cohort. **b** Output of the cross-validation search of algorithms, hyperparameters, and the number of proteins is displayed as the area under the receiver operating characteristic curve (AUROC) and Matthews Correlation Coefficient (MCC) from the predictions on the test sets. The number of proteins used in each model is shown to the right. The error bars represent the standard deviation across the 5-fold cross-validation runs. **c** Recursive feature elimination for the best-performing algorithm, the support vector classification (SVC) model, is illustrated. The x-axis represents the number of proteins, and the y-

axis the weighted F1-score. The SVC model achieved high performance across the entire range of protein numbers, with the highest performance obtained using four proteins. **d** Boxplots of the probability of each sample being classified as MIS-C was computed using the support vector classification models from each cross-validation. The probabilities were plotted for the test set and for each sample in the different diagnostic groups. The inner quartile range (IQR) is represented by a box, the median as a line in the box and 1.5xIQR as whiskers. **e** Boxplots of the protein levels across the different diagnostic groups (measured in label-free quantification) for the four proteins included in the diagnostic signature. Fc Gamma Receptor IIIa (FCGR3A), lymphocyte cytosolic protein 1 (LCP1), alpha-1-antichymotrypsin (SERPINA3) and butyrylcholinesterase (BCHE). The inner quartile range (IQR) is represented by a box, the median as a line in the box and 1.5xIQR as whiskers. **f** Prediction performance of the 4-protein diagnostic signature is depicted with the area under the receiver operating characteristic curve (AUC) and Matthews Correlation Coefficient (MCC). Standard deviation from the 5 cross-validation folds is shown as semi-transparent error borders on the curves. The shadings represent the standard deviation in tpr/fpr values across the 5-fold cross-validation. The text in the plot "AUC = 0.87 (std = 0.1)" refers to the variation in AUC across the 5-fold cross-validation runs.

25 healthy controls (PXD029375)[37]. Data were z-scored (sample-wise) and only proteins found in both cohorts were used as features in the model. We used the same algorithm type and hyperparameters as the previous model, but the model was refitted with the new input consisting of the protein overlap between the two cohorts. The model was trained on our training set and applied to the test set and the external validation cohort.

The performance metrics used included the MCC, AUC, ROC curves, the distribution of prediction probabilities, and confusion matrices.

### Reporting summary
Further information on research design is available in the Nature Portfolio Reporting Summary linked to this article.

### Data availability
The mass spectrometry proteomics data have been deposited to the ProteomeXchange Consortium via the Proteomics Identifications Database (PRIDE) partner repository[38], with the dataset identifier PXD045661. The source data behind the graphs in the paper can be found in Supplementary Data 1-4.

### Code availability
The jupyter notebooks are available at https://github.com/nicwin98/MIS-C.

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

## Acknowledgements

We thank Christine Rasmussen (Department of Clinical Biochemistry, Copenhagen University Hospital, Bispebjerg) for her time planning, preparing and the proteomic analysis and we also acknowledge the Clinical Proteomic Group at the NNF Center for Protein Research, University of Copenhagen. Funding was received from the Novo Nordisk Foundation (grant NNF19OC0055001) to Nicolai Jacob Wewer Albrechtsen supporting LD, MO, and CR, Novo Nordisk Foundation Center for Protein Research is supported financially by the Novo Nordisk Foundation (Grant agreement NNF14CC0001). Funding was received from the National Ministry of Higher Education and Science (grant 0237–00004B), Innovation Fund Denmark (0176–00020B), Greater Copenhagen Health Science Partners (CAG CHILD) to Kjeld Schmiegelow, Ulrikka Nygaard, and Nadja Hawwa Vissing supporting KHSD, LSS, and MH.

## Author contributions

UN, Annelaura Bach Nielsen (ABN), KHSD, LD, MH, Allan Nielsen (AN), JPG, LSS, DC, IMJ, KS, NHV and NJWA conceptualized this nationwide study. UN, KS, NHV, MM and NJWA obtained funding for the study. UN, KHSD, MH, AN, JPG, NHV and LSS obtained clinical details for patients with MIS-C through electronic medical records. UN, ABN, LD, KHSD, NHV and NJWA had full access to all the data in the study and take responsibility for the integrity of the data and accuracy of the data analysis. UN, ABN, LD, MEO, THM, KHSD, NHV and NJWA analyzed data. UN, ABN, KHSD, NHV and NJWA drafted the first version of the manuscript. All authors contributed to the data interpretation. All authors revised the manuscript critically for important intellectual content. All authors finally approved the work.

## Competing interests

The authors declare no competing interests.
