## [Peer Review File · Communications Biology]

Reviewers' comments:

Reviewer #1 (Remarks to the Author):

Thank you for the opportunity to review this interesting study.

In the submitted manuscript, the authors reported that using mass-spectrometry proteomics and AI-assisted unbiased analysis they identified four plasma protein biomarkers (LCP1, FCGR3A, SERPINA3, BCHE) that could highly discriminate multisystem inflammatory syndrome in children (MIS-C) from Kawasaki disease (KD) and severe bacterial and viral infections. They used MIS-C patients meeting CDC case definitions and relevant control groups. I believe the manuscript adds to foundation for the development of a diagnostic test that could assist pediatricians to timely diagnose MIS-C. However, I have some questions I hope the authors could address:

Major Comments:

- While the authors used relevant control groups to compare with MIS-C cases, one major limitation of this study is the small sample size, particularly in KD patients, limiting further validation of results observed in this study and this is briefly mentioned in the Discussion. Sample size calculations are not provided in the methods sections to determine the required sample size for each diagnostic category included in this study.

Another major limitation was the lack of validation of the mass spectroscopy results by antibody-based methods that are more amenable to translation into a diagnostic test. This was not mentioned as a limitation, only the fact they were unable to validate the results in an independent cohort.

Major points:

1. Did the authors compare the performance of their signature in MIS-C cases arising from the Alpha vs Delta variant?
2. Given the large proteomic dataset that was generated in this study, did the authors also look for markers of severity in MIS-C cases (as many were severe with shock, requiring ICU and/or inotropes) or only diagnostic markers?
3. Lines 146-149: 'All 12 machine-learning algorithms, except one, had MCC and AUC above 0.77 (Figure 3B). We continued the subsequent analysis with a support vector classification model that, during model training, had an AUC and MCC of 100% and 1, respectively.' It's not clear what was done here in terms of improving the AUC by 23%. This needs explained in more detail and confidence intervals added.
4. Lines 181-182: 'The 4-protein diagnostic signature is the first proteomic-based diagnostic signature for MIS-C' I refer the author to this paper:
<https://www.medrxiv.org/content/10.1101/2023.07.28.23293197v1.full>
5. A possible role for 3 of the 4 proteins in the diagnostic signature in MIS-C pathogenesis is not mentioned and needs added to the Discussion.

Minor points:

1. Table 1 provides basic clinical information. This could be improved by including data on oxygen requirements, PICU and inotropes and also clinical features including GI involvement, cardiac, shock and comorbidities for the MIS-C cases and febrile controls. The authors comment on the bacteria and sepsis patients in Table 1 but don't mention those with viral infection.
2. Line 108: '66 proteins related to therapy with intravenous immunoglobulin were excluded'. How were

these proteins identified? This should be referenced.

3. Lines 110-111: Typo as this related to Figure 2 A/B. The correlation coefficient not stated or a p-value in relation to the 105 significant proteins (Line 112).

4. Lines 145-146: What about the KD patients? Only viral and bacterial patients are mentioned.

5. Lines 150-153: The function of 2/4 of the proteins in the signature is mentioned here. I would recommend to either include the role for all four or save this for the Discussion.

6. Lines 207-208: Could this also mean these proteins are highly correlated with other proteins significant between MIS-C and febrile controls?

7. Lines 223-224: 'We found profound changes in lipid metabolism, previously unreported in MIS-C.' I refer the author to these papers:

<https://www.ncbi.nlm.nih.gov/pmc/articles/PMC8624489/>

<https://www.frontiersin.org/articles/10.3389/fcvm.2022.1033660/full>

8. Lines 269-271: How were the bacterial and viral infections defined/ phenotyped? Any deaths in the MIS-C or febrile control groups?

9. Lines 314-315: What about KD cases? Where they also included? They cannot be considered viral or bacterial.

Reviewer #3 (Remarks to the Author):

The authors utilized AI-assisted proteomics to characterize a novel 4-protein signature and 7 protein clusters from children with MIS-C compared to those with non-MIS-C febrile illnesses (bacterial, viral, severe sepsis, or KD) enrolled prior to the start of the COVID-19 pandemic. The authors utilized a training and validation internal set of data to develop their model and then validated it using an external cohort from a proteomics database to identify four specific proteins- lymphocyte cytosolic protein 1, FcγRIIIa, alpha-1-antitrypsin, and butyrylcholinesterase, which have the potential to be utilized for novel diagnostic tests. The study is important in that we still do not have a diagnostic test for MIS-C, nor do we know the exact pathogenesis. There is often overlap in the clinical definition between MIS-C and other hyperinflammatory syndromes such as Kawasaki Disease, Toxic shock syndrome, rheumatologic conditions, or severe COVID-19, making the diagnosis and thus optimal management of MIS-C challenging. One concern I have that the authors also highlight in their limitations is the choice for controls and lack of an external validation cohort. Routine bacterial and viral infections most often do not meet the clinical criteria for MIS-C, and thus having a biomarker to distinguish MIS-C from these entities may not be the most clinically useful. For example in Table 1, the viral cohort had a median CRP of 5 whereas those with acute MIS-C had a median CRP of 219. Instead, having a biomarker to distinguish MIS-C from entities with similar clinical presentations, such as severe COVID-19 or Kawasaki disease (which had a median prediction probability of 55.8% in the validation of the 4-protein signature), would in fact be clinically meaningful, particularly as the treatments vary and are time sensitive. Furthermore, it would certainly strengthen the study and increase generalizability of results if the

authors are able to validate their 4-protein diagnostic signature in an external, blinded cohort, as a limitation of large amounts of data can be a bias towards significance. Nevertheless, the results are interesting and advance our knowledge of MIS-C. Few minor comments:

There have been a handful of papers published on the use of proteomics in MIS-C (<https://doi.org/10.1172/JCI151520>, <https://doi.org/10.1038/s41467-021-27544-6>) which the authors may want to include. In the JCI paper by Porritt RA et al, authors utilized proteomics to distinguish severe MIS-C from mild disease and KD, and similarly found upregulation in FcGR11a and a reduction in proteins involved with lipid metabolic processes, lipoprotein clearance, and components of the coagulation cascade.

Could the authors provide a reference for the external US validation cohort of MIS-C and controls patients?

Rebuttal Letter to the Reviewers

Manuscript reference number: COMMSBIO-23-4346-T.

Manuscript title: Proteomic profiling of Multisystem Inflammatory Syndrome in Children: Exploring Pathogenesis and Diagnostic Signatures

Reviewer #1 (Remarks to the Author):

Thank you for the opportunity to review this interesting study.

In the submitted manuscript, the authors reported that using mass-spectrometry proteomics and AI-assisted unbiased analysis they identified four plasma protein biomarkers (LCP1, FCGR3A, SERPINA3, BCHE) that could highly discriminate multisystem inflammatory syndrome in children (MIS-C) from Kawasaki disease (KD) and severe bacterial and viral infections. They used MIS-C patients meeting CDC case definitions and relevant control groups. I believe the manuscript adds to foundation for the development of a diagnostic test that could assist pediatricians to timely diagnose MIS-C.

Response:

Thank you very much for reviewing this paper!

However, I have some questions I hope the authors could address:

Major Comments:

- While the authors used relevant control groups to compare with MIS-C cases, one major limitation of this study is the small sample size, particularly in KD patients, limiting further validation of results observed in this study and this is briefly mentioned in the Discussion.

Response:

We agree and acknowledge that the number of KD patients in this study is very small. Before the initiation of the study, we considered omitting KD patients due to the low number (all identified prior to the pandemic). However, we decided to include them to illustrate the performance of the signature on KD patients, despite the few.

- Sample size calculations are not provided in the methods sections to determine the required sample size for each diagnostic category included in this study.

Response:

We did not perform sample size calculations as the nature of the study was exploratory. However, we found numerous significant changes (also corrected for multiple testing) based on a relatively small number of participants. This supports that sample size per se was not problematic but we agree with the reviewer that a larger sample size may have provided additional insight and lower p-values. We have now inserted this in the method section in the revised manuscript.

Change in methods section:

Sample size calculations were not performed due to the exploratory nature of the study.

- Another major limitation was the lack of validation of the mass spectroscopy results by antibody-based methods that are more amenable to translation into a diagnostic test. This was not mentioned as a limitation, only the fact they were unable to validate the results in an independent cohort.

Response:

Mass-spectrometry was considered a more specific measurement technique than antibody-based methods. This is due to the fact that antibodies may suffer from interference due to their epitope specificity and matrix effects (e.g., plasma moieties). We believe the presented data thereby offer a more specific approach to measure proteins as also generally known as 'gold standard' for protein chemistry.

Major points:

1. Did the authors compare the performance of their signature in MIS-C cases arising from the Alpha vs Delta variant?

Response:

Thank you for this question. We did not compare the performance on the 4-protein signature on MIS-C cases arising from the Alpha vs. Delta variant due to the overall low number in our study. In our cohort, we have previously shown that the clinical phenotype was similar between the two variants (Holm, 2022, JAMA Ped; doi:10.1001/jamapediatrics.2022.2206

2. Given the large proteomic dataset that was generated in this study, did the authors also look for markers of severity in MIS-C cases (as many were severe with shock, requiring ICU and/or inotropes) or only diagnostic markers?

Response:

This is a very relevant question. Thank you. As most of our included patients 21 (78%) presented with shock, we did not look for markers of severity markers. Thus, the 4-protein signature primarily reflects children with severe MIS-C.

3. Lines 146-149: 'All 12 machine-learning algorithms, except one, had MCC and AUC above 0.77 (Figure 3B). We continued the subsequent analysis with a support vector classification model that, during model training, had an AUC and MCC of 100% and 1, respectively.' It's not clear what was done here in terms of improving the AUC by 23%. These needs explained in more detail and confidence intervals added.

Response

We apologize for not being clear on this part. We evaluated 12 machine-learning algorithms. Eleven algorithms had MCC and AUC between 0.77 and 1, while one (QDA) had MCC and 0.27 and AUC of 0.65. Figure 3B shows the performance of 12 different algorithms. We did not do anything to improve the algorithms, but chose the best model (SVC), which had MCC and AUC of 1. All 12 algorithms were trained on the same data, so the difference in performance depended on how well each algorithm modelled the data. The SVC algorithm had fully learned the input data, whereas the GP algorithm had partially learned the data structure, which can be seen by the 23% lower MCC on the same data. Thus, we did not further improve the models. Confidence intervals (standard deviations) are illustrated as errorbars on the barplot (Figure 3B). We have now further elaborated this in the results section to make it clear for the reader.

Diagnostic classifier for MIS-C using machine learning

The data set used to develop a diagnostic signature for MIS-C included children with MIS-C and febrile controls with viral and bacterial infections (Figure 3A). All 12 machine-learning algorithms, except one, had MCC and AUC ~~between above~~ 0.77-1 (Figure 3B). We continued the subsequent analysis with ~~the a~~-support vector classification (SVC) model that, ~~during model training,~~ had an AUC and MCC of 100% and 1, respectively (Figure 3B). Recursive feature elimination revealed

4. Lines 181-182:: 'The 4-protein diagnostic signature is the first proteomic-based diagnostic signature for

MIS-C' I refer the author to this paper:

<https://www.medrxiv.org/content/10.1101/2023.07.28.23293197v1.full>

Response

We apologize for not identifying this paper in preprint. We have now changed the wording in the manuscript and included this reference in the paper. We have rewritten the discussion section to include this paper.

signature with an AUC of 100% based on only four proteins. The 4-protein diagnostic signature ~~is the first unbiased? proteomic based diagnostic signature for MIS-C [Yeoh et al., preprint].~~ It holds promising avenues for developing a rapid, and low-cost, diagnostic bedside test with important

different proteins, had high AUCs. This supports that several proteins may be used for an MIS-C signature and emphasizes proteomics as a very powerful tool for MIS-C diagnostics.

~~Previously, a 3-protein signature has been shown to distinguish MIS-C patients from other disease controls with an AUC of 86% in a study investigating seven host proteins [ref].~~

~~Additionally,~~ Recently, a diagnostic signature based on a 5-gene blood RNA expression signature for MIS-C ~~has been~~was reported.²² It was validated with an RT-qPCR assay and revealed a high diagnostic accuracy. ~~Collectively, These results~~ suggests that MIS-C has distinct host responses detectable by both transcriptomics and proteomics, which may be suitable for innovative diagnostic tests. However, a diagnostic test based on ~~four few~~ plasma proteins may reveal a faster result and may be cheaper to implement in clinical medicine as a routine laboratory analysis.

5. A possible role for 3 of the 4 proteins in the diagnostic signature in MIS-C pathogenesis is not mentioned and needs added to the Discussion.

Response

Thank you for this suggestion. We have added roles for all four proteins in the discussion section.

previous studies.²³⁻²⁵ Consistently, three of the four proteins in the identified 4-protein signature, were involved immune dysregulation: Alpha-1-antichymotrypsin is involved in complement activation, while Fc Gamma Receptor IIIa is involved in antibody-dependent cellular toxicity. Further, Notably, the increased levels of lymphocyte cytosolic protein 1, which was elevated and beta-2-microglobulin indicated activation of T-cells, supporting the hypothesis of a super-antigen-mediated polyclonal T-cell activation.²⁶ The last protein in the 4-protein signature was butyrylcholinesterase, which hydrolyses choline esters and was reduced. The significance of this protein in the pathogenesis of MIS-C is unknown.

Minor points:

1. Table 1 provides basic clinical information. This could be improved by including data on oxygen requirements, PICU and inotropes and also clinical features including GI involvement, cardiac, shock and comorbidities for the MIS-C cases and febrile controls. The authors comment on the bacteria and sepsis patients in Table 1 but don't mention those with viral infection.

Response

Thank you for these suggestions. We have now revised Table 1 and provided additional data, including PICU, inotropes, and organ manifestations. In the table and results section, we have added details on patients with viral infections.

collection (day ⁻¹)		
Clinical characteristics⁴		
Hypotension	21 (78%)	0
Cardiac involvement	27 (100%)	-
Gastrointestinal involvement	25 (93%)	-
Hematologic involvement	27 (100%)	-
Dermatologic involvement	25 (93%)	-
Neurologic involvement	0	-
Respiratory involvement	7 (26%)	-
Renal involvement	13 (48%)	-
C-reactive protein (mg/L, max)	219 (84-402)	87 (83-202)
Treatment		
Antibiotic therapy	26 (96%)	-
Inotropes	9 (33%)	
Immunoglobulins	24 (89%)	-
Glucocorticoids	24 (89%)	-
Anakinra	10 (37%)	-
Intensive Care Unit	24 (89%)	
Length of hospital stay (days, range)	6 (3-13)	-

² Viral controls all had fever, proven viral detection in nasopharyngeal specimen, and C-reactive protein below 25 mg/L

2. Line 108: '66 proteins related to therapy with intravenous immunoglobulin were excluded'. How were these proteins identified? This should be referenced.

Response

Immunoglobulin proteins excluded were proteins from the heavy-chains, light-chains, j-chains and variable regions. This is now stated in the manuscript in the method section

Mass spectrometry data analysis

Venous blood samples were collected into EDTA-containing tubes, spun at 3,000g for 10min at 4°C within 2 hours, and stored at -80°C. Sample preparation for proteomic analysis was performed as previously published.¹⁶ Samples were analyzed using an Exploris 480 Thermo Fischer Scientific system by Eyosep One (Eyosep Biosystem, Denmark) and proteomic data were acquired in a data-independent acquisition mode. Proteins related to therapy with intravenous immunoglobulin, including heavy-chains, light-chains, j-chains and variable regions, were excluded (ref). The mass spectrometry raw files were processed with Spectronaut version 17 (Biognosys, Zurich, Switzerland). A previously generated plasma spectral library containing 2137 protein groups and 16,254 peptides was used.

3. Lines 110-11: Typo as this related to Figure 2 A/B. The correlation coefficient not stated or a p-value in relation to the 105 significant proteins (Line 112).

Responses

Thank you for identifying these typos. They have been corrected.

visualized by the UMAP plot (Figure 42A) and the unsupervised heatmap (Figure 42B), which both

The p-values are provided in sTable2.

Correlation coefficients between each sample (94 samples x 94 samples) are illustrated in 2B and colored according to their Pearson correlations (from dark red to white corresponding to Pearson correlations from 0.75 to 1)

4. Lines 145-146: What about the KD patients? Only viral and bacterial patients are mentioned.

Response

We apologize for not being clear on the matter. In the initial proteomics analyses on the pathogenesis, MIS-C was compared to febrile controls, including viral and bacterial infections, as well as KD and severe sepsis.

We excluded KD and severe sepsis in the development of a diagnostic signature in order to use these two groups as validation cohorts. This has now been rewritten in the manuscript to make it clear for the reader.

Result section:

Plasma proteins in children with MIS-C compared to controls

The data set used to identify differences in protein levels included patients with MIS-C, febrile controls with viral and bacterial infections, KD, and severe sepsis (Figure 1A). We identified 450

Diagnostic classifier for MIS-C using machine learning

The data set used to develop a diagnostic signature for MIS-C included children with MIS-C and febrile controls with viral and bacterial infections (Figure 3A). All 12 machine-learning algorithms,

5. Lines 150-153: The function of 2/4 of the proteins in the signature is mentioned here. I would recommend to either include the role for all four or save this for the Discussion.

Response

Thank you for this suggestion. We have deleted the function in the results section and added roles for all four proteins in the discussion section.

antichymotrypsin, involved in the immune response, and butyrylcholinesterase, which hydrolyzes choline esters (Table 2; Figure 3E).

previous studies.²³⁻²⁵ Consistently, three of the four proteins in the identified 4-protein signature, were involved immune dysregulation: Alpha-1-antichymotrypsin is involved in complement activation, while Fc Gamma Receptor IIIa is involved in antibody-dependent cellular toxicity. Further, Notably, the increased levels of lymphocyte cytosolic protein 1, which was elevated, and beta-2-microglobulin indicated activation of T-cells, supporting the hypothesis of a super-antigen-mediated polyclonal T-cell activation.²⁶ The last protein in the 4-protein signature was butyrylcholinesterase, which hydrolyses choline esters and was reduced. The significance of this protein in the pathogenesis of MIS-C is unknown.

6. Lines 207-208: Could this also mean these proteins are highly correlated with other proteins significant between MIS-C and febrile controls?

Response

We agree that a protein signature using different proteins could have been a possibility since a different support vector classification model with different proteins also achieved a high diagnostic accuracy of MIS-C. This supports that several proteins may be used for an MIS-C signature and emphasizes the robustness of proteomics as a diagnostic tool for MIS-C.

7. Lines 223-224: 'We found profound changes in lipid metabolism, previously unreported in MIS-C.' I refer the author to these papers:

<https://www.ncbi.nlm.nih.gov/pmc/articles/PMC8624489/>

<https://www.frontiersin.org/articles/10.3389/fcvm.2022.1033660/full>

Response

Thank you for this suggestion. We have added these papers in the discussion section.

We found profound changes in lipid metabolism, consistent with previous studies, [Mietus-Snyder et al., Verduci et al.] previously unreported in MIS-C. Lipid mediators have been described to be

8. Lines 269-271: How were the bacterial and viral infections defined/ phenotyped? Any deaths in the MIS-C or febrile control groups?

Response

Details of the controls with bacterial and viral infection are now added to the manuscript.

Results section:

*collected when fully recovered a median of 39 days (19-76) following the diagnosis of MIS-C. All bacterial febrile controls had urinary tract infection confirmed by positive urine dipstick and urine culture with *Escherichia coli*. Viral controls all had fever, proven viral detection in nasopharyngeal specimen, and C-reactive protein below 25 mg/L (Table 1). Characteristics of the febrile controls are summarized in Table 1. There were no deaths in any of the groups.*

Table 1

¹ Bacterial infection included children with proven urinary tract infections

² Viral controls all had fever, proven viral detection in nasopharyngeal specimen, and C-reactive protein below 25 mg/L

9. Lines 314-315: What about KD cases? Where they also included? They cannot be considered viral or bacterial.

Response

We apologize for not being clear on the matter. As also described above, in the initial proteomics analyses on the pathogenesis, MIS-C was compared to febrile controls, including viral and bacterial infections, as well as KD and septic shock. We excluded KD and septic shock in the development of a diagnostic signature in order to use these two groups as validation cohorts. This has now been rewritten in the manuscript to make it clear for the reader.

Result section:

Plasma proteins in children with MIS-C compared to controls

The data set used to identify differences in protein levels included patients with MIS-C, febrile controls with viral and bacterial infections, KD, and severe sepsis (Figure 1A). We identified 450

Diagnostic classifier for MIS-C using machine learning

The data set used to develop a diagnostic signature for MIS-C included children with MIS-C and febrile controls with viral and bacterial infections (Figure 3A). All 12 machine-learning algorithms,

Reviewer #3 (Remarks to the Author):

The authors utilized AI-assisted proteomics to characterize a novel 4-protein signature and 7 protein clusters from children with MIS-C compared to those with non-MIS-C febrile illnesses (bacterial, viral, severe sepsis, or KD) enrolled prior to the start of the COVID-19 pandemic. The authors utilized a training and validation internal set of data to develop their model and then validated it using an external cohort from a proteomics database to identify four specific proteins- lymphocyte cytosolic protein 1, FcγRIIIa, alpha-1-antitrypsin, and butyrylcholinesterase, which have the potential to be utilized for novel diagnostic tests. The study is important in that we still do not have a diagnostic test for MIS-C, nor do we know the exact pathogenesis. There is often overlap in the clinical definition between MIS-C and other hyperinflammatory syndromes such as Kawasaki Disease, Toxic shock syndrome, rheumatologic conditions, or severe COVID-19, making the diagnosis and thus optimal management of MIS-C challenging. One concern I have that the authors also highlight in their limitations is the choice for controls and lack of an external validation cohort. Routine bacterial and viral infections most often do not meet the clinical criteria for MIS-C, and thus having a biomarker to distinguish MIS-C from these entities may not be the most clinically useful. For example in Table 1, the viral cohort had a median CRP of 5 whereas those with acute MIS-C had a median CRP of 219. Instead, having a biomarker to distinguish MIS-C from entities with similar clinical presentations, such as severe COVID-19 or Kawasaki disease (which had a median prediction probability of 55.8% in the validation of the 4-protein signature), would in fact be clinically meaningful, particularly as the treatments vary and are time sensitive.

Furthermore, it would certainly strengthen the study and increase generalizability of results if the authors are able to validate their 4-protein diagnostic signature in an external, blinded cohort, as a limitation of large amounts of data can be a bias towards significance.

Response

Thank you very much for reviewing this paper. We agree that it would have improved the paper significantly if we could have validated the 4-protein signature in an external, blinded cohort. However, we were not able to identify any studies, which have performed unbiased mass-spectrometry proteomics or any studies who have investigated the four proteins in our signature. Further, the decline in the incidence of MIS-C has refrained us from validating the accuracy of the 4-protein signature in a prospective Danish MIS-C cohort.

Nevertheless, the results are interesting and advance our knowledge of MIS-C. Few minor comments:

There have been a handful of papers published on the use of proteomics in MIS-C

(<https://doi.org/10.1172/JCI151520>, <https://doi.org/10.1038/s41467-021-27544-6>) which the authors may want to include. In the JCI paper by Porritt RA et al, authors utilized proteomics to distinguish severe MIS-C from mild disease and KD, and similarly found upregulation in FcGR111a and a reduction in proteins involved with lipid metabolic processes, lipoprotein clearance, and components of the coagulation cascade.

Response

Thank you for these relevant papers, including the paper finding an upregulation in FcGR111a. They are now added to the manuscript in the discussion section

Overall Our results align with those of complement a previous study, which also reported upregulated Fc Gamma Receptor 111a, immune and complement activation and reduced lipid transport and clearance mechanisms (5). Further, our results were consistent with studies
exploring single coagulation factors or coagulation profiles.^{31,32} The significant alterations in

Could the authors provide a reference for the external US validation cohort of MIS-C and controls patients?

Response

Yes, we excuse. This reference has now been added to the manuscript

REVIEWERS' COMMENTS:

Reviewer #1 (Remarks to the Author):

I have read the revised manuscript and am satisfied with the responses you provided, the additional text and corrections you have made to the manuscript and to the clinical table. I recommend that this paper can now be accepted.

Reviewer #3 (Remarks to the Author):

The authors have made adequate effort at addressing the reviewers concerns and comments within the limits of their cohort.